# Impact of the entorhinal feed-forward connection to the CA3 on hippocampal coding

**Samuel B. Lassers**[1⦿], **Shazfa S. Khatri**[1⦿], **Ruiyi Chen**[1], **Yash S. Vakilna**[1,2],
**William C. Tang**[1], **Gregory J. Brewer**[1,3]*

**1** Department of Biomedical Engineering, University of California Irvine, Irvine, California, United States of America, **2** Texas Institute of Restorative Neurotechnologies (TIRN), The University of Texas Health Science Center (UTHealth), Houston, Texas, United States of America, **3** Memory Impairments and Neurological Disorders (MIND) Institute, Center for Neuroscience of Learning and Memory, University of California Irvine, Irvine, California, United States of America

⦿ These authors contributed equally.

* GJBrewer@uci.edu

## Abstract

Each sub-region of the hippocampus plays a critical computational role in the formation of episodic learning and memory, but studies have yet to show and interpret the individual spiking dynamics of each region and how that information is passed between each subregion. This is in part due to the difficulty in accessing individual communicating axons. Here, we created a novel microfluidic device that facilitates network growth of four separated hippocampal subregions over a micro-electrode array. This device enabled monitoring single axons over two electrodes so direction of spike propagation in interregional communication could be ascertained. In this *in vitro* hippocampal study, we compared spiking dynamics across two novel four-compartment device architectures: one with four sets of axon tunnels between subregions that excluded the perforant pathway from EC-CA3, and one with five sets of axon tunnels that included the EC-CA3 connection. We found 30–90% faster feed-forward firing rates (shorter interspike intervals) in axons in the five-tunnel model with 35–75% slower bursting dynamics (longer interburst intervals) compared to the four-tunnel model. The CA3-CA1 and CA1-EC axons had more spikes in bursts in the five-tunnel architecture than the four-tunnel counterpart suggesting more structured information transfer. Feedback firing rates were similar between configurations. The faster feed-forward inter-regional spiking in the more natural five-tunnel than the four-tunnel configuration suggests tighter control of spiking and possibly more precise communication between subregions.

## Introduction

The flow of information through the hippocampal trisynaptic circuit enables learning and the formation of episodic memory. The anatomical structure of this pathway,

**Data availability statement:** The data is available from https://datadryad.org/dataset/doi:10.5061/dryad.x0k6djhwb.

**Funding:** SBL was supported in part by funds from the UC Irvine Foundation. The Foundation had no role in study design, data collection and analysis, decision to publish or preparation of the manuscript. There were no additional external funding received for this study. SSK, RC, YSK, WCT, and GJB did not receive additional external funding for this study.

**Competing interests:** The authors declare that the research was conducted in the absence of any commercial or financial relationships that could be construed as a potential conflict of interest.

initially detailed by Ramon y Cajal in 1911, reveals a sequential relay of connections among its three regions, each with its own computational role. Information from other regions of the brain is integrated in the entorhinal cortex (EC), progresses to the dentate gyrus (DG), passes through CA3 and then CA1 [1] before completing the loop by returning to the subiculum and entorhinal cortex. Later studies clearly identified the perforant path that connects the EC to CA3 and CA1 as well as DG, a feed-forward jump in connectivity of unclear function. Although the functions of the hippocampal subregions are well understood, the firing dynamics in CA3 in response to DG and EC inputs have not been sufficiently explored in part because of a lack of access to the afferent axons.

The entorhinal cortex (EC) integrates sensory information from the prefrontal cortex, amygdala, and the orbitofrontal cortex allowing for further computation of object and spatial representations and reward related information [2]. Primary inputs to the DG from the EC via the perforant pathway are proposed to be essential for pattern separation [2] and mossy fiber connections from the DG synapse in the CA3, facilitating the learning of distinctive and novel features. The CA3 associates diverse types of information and excels in the rapid encoding of novel information while aiding in efficient recall processes of pattern completion [3,4]. More activity in CA3 has also been shown to support discrimination between stimuli that share features [5]. The CA1 sequences and orders information coming from the CA3 and then routes information back to the EC and subiculum. While the separate functions of the subregions of the trisynaptic are well understood, a thorough understanding of effects on CA1 spiking dynamics due to the CA3 inputs from DG compared to both the EC and DG inputs to CA3 could provide important information for modeling architecture as well as hippocampal coding functions. This study aims to investigate how the absence and presence of the EC-CA3 perforant pathway affects the firing dynamics and interregional communication in the trisynaptic loop using a novel microfluidic device design.

In this study we examined the spiking dynamics across the hippocampus through the comparison of two novel four-compartment devices: one with four sets of axon tunnels between subregions monitored by substrate electrodes that excluded the perforant pathway from EC-CA3, and one with five sets of monitored axon tunnels that included the EC-CA3 connection. The narrow tunnels allowed the isolation of single axons for unambiguous directionality and high impedance measures of single action potentials. Using our microelectromechanical systems (MEMS) device, we simultaneously recorded spiking activity in the four subregions of the hippocampus and the axons connecting the sub-regions. The directionality of spike propagation was measured using electrode pairs underneath single axons between sub-regions to determine their feed-forward or feedback activity. In our previous study [6], we demonstrated spontaneous directional spatiotemporal dynamics in soma and axons with the four-tunnel architecture. We saw classical log-log subregion specific distributed dynamics of interspike intervals (ISIs) and interburst intervals (IBIs) typically found in neural networks [7]. This provided a solid basis for the exploration of how the additional tunnels connecting EC to CA3 affects network dynamics. In the present

study, we expanded these findings by comparing these spikes and burst distributions in the monitored four-tunnel sets with those of five-tunnel set model of the hippocampus to explore how the perforant path of EC-CA3 axons might affect network dynamics. For the first time, we report comparisons of explicit axonal communication between modeled hippocampal subregions with determined directionality. This could provide important insights for modeling layer skipping in complex network architectures and the subsequent effects on hippocampal coding functions.

## Materials and methods

### In vitro hippocampal neuronal network culture in a four-chamber device with four-way and five-way microfluidic interconnections

Microfluidic and culturing methods were previously described in more detail in Vakilna et al. [8,9]). In short, we employed a novel four-chamber device containing microfluidic tunnels for axonal communication between chambers (Fig 1). Each chamber contained dissociated neurons from micro dissections of the entorhinal cortex (EC), dentate gyrus and hilus (DG), CA3, and CA1 including the subiculum from postnatal day 4 Sprague Dawley rat pups under anesthesia as approved by the UC Irvine Institutional Animal Care and Use Committee (IACUC). Neonatal mice were exposed to an analgesic ice bath to minimize suffering and then sacrificed by decapitation. Brain cells were dissociated and plated at 1,000 cells/mm$^2$ for DG, 330 for CA3, 410 for CA1 (including subiculum), and 330 for EC, in order to reflect in vivo neural densities: EC-DG 1:3, DG-CA3 3:1, CA3-CA1 1:1.25, and CA1-EC 1.25:1 which are ratios consistent with previous *in vivo* findings [10–14]. Cells were in 10 μL of NbActiv4 medium (Transnetyx BrainBits, Springfield, IL [15] and were plated into the wells sequentially. After thirty minutes in the incubator to allow for adhesion, 0.8 mL culture medium was added. The cultures were capped with Teflon sheets (ALA Scientific, Farmingdale, NY) and incubated for 21–26 days in humidified 5% $CO_2$ and 9% $O_2$ [16]. Half of the medium was changed every 3–4 days. Activity was recorded on days 21–26, 2–5 days after a medium change when the networks had reached maturity. MEA120 glass multielectrode arrays (MEA) equipped with 120 30-μm-diameter electrodes spaced 200 μm served as the substrates for the culture of neuronal networks (Multichannel Systems, Reutlingen, Germany; ALA Scientific, Farmingdale, NY, USA). A custom polydimethylsiloxane (PDMS) device was aligned and attached to the MEA that separates the microelectrodes into four chambers for each cell type. Axon-isolating tunnels connected each compartment to form features of the trisynaptic loop. Each PDMS well was 9.7 mm$^2$ by 1-mm high. Each microfluidic tunnel was 3 μm high × 10 μm wide × 400 μm long spaced 50 μm apart. The self-wiring nature of the dissociated neuronal cells allows axons to grow through these tunnels. The dimensions of the tunnels are optimized for single axons whose direction of information transmission is captured by the delay in spiking activity across two electrodes in the tunnel.

A) Four-tunnel architecture

B) Five-tunnel architecture

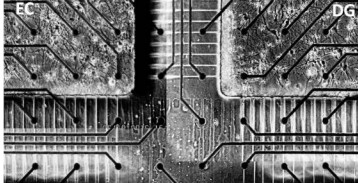
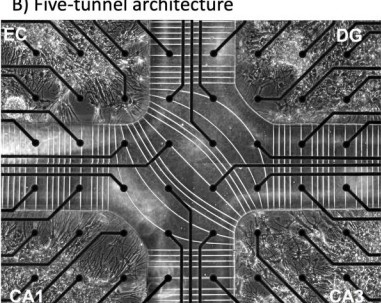

**Fig 1. Comparison of four- versus five-tunnel architecture.** (A) Four sets of tunnels connect EC to only DG and CA1 and CA3 to CA1. (B) Five sets of tunnels add cross tunnels from EC to CA3. White lines are microfluidic tunnels for axonal communication between labeled subregions that confine neuronal somata that were micro-dissected from individual subregions of the postnatal hippocampus. Dark circles are 30 μm diameter electrodes used to measure voltage spikes. Inter-electrode spacing is 200 μm. Black lines are insulated connections to the amplifier. Adapted from Wang et al. [17]) and Chen et al. [9].

Two configurations of microfluidic tunnels for axonal communication were compared (Fig 1). In the initial four-tunnel set configuration of the microfluidic tunnels (Fig 1A), each compartment was connected to the adjacent one to allow feed-forward connection according to Cajal's original understanding of the trisynaptic loop that omits the EC-CA3 connection. Five of the 51 axon tunnels between each compartment were monitored with a pair of electrodes underneath. In the second five-tunnel set configuration, two electrodes were under each of 4 of the 67 tunnels between the same subregions, but to instantiate the perforant path [1], axonal tunnels diagonally connected the EC to the CA3 wells. Five of these 11 EC to CA3 tunnels were monitored by pairs of electrodes and two more by a single electrode (Fig 1). From the spike timing delay between the electrode pairs in each tunnel, we determined the direction of axonal potential propagation and categorization of action potentials as either feed-forward or feedback. In both configurations, nineteen electrodes were at the bottom of each subregion. The substrate was treated with oxygen plasma followed by coating the wells with poly-D-lysine for cell adhesion. Finally, we placed micro dissected dissociated neurons from four-day-old rat hippocampi in the corresponding wells and cultured them in NbActive4 [18] for 18–22 days [9].

## Multi-electrode array and recording

A Multichannel Systems MEA120 1100 (Multichannel Systems, Reutlingen, Germany) amplifier was used to record spontaneous activity on the 120-electrode microarray. Spontaneous activity was analyzed using MC_Rack software at a sampling rate of 25 kHz at 37° C for 5 min, in humidified 5% $CO_2$, 9% $O_2$ (custom Airgas USA, Santa Ana, CA). Recordings were initiated several minutes after transfer from the culture incubator to the amplifier, once at least 80% of the tunnels had stable activity. Arrays with <80% active tunnels or that had poor growth in one of the compartments were rejected for recording.

## Spike detection, sorting, and axonal propagation direction

Details on our axonal spike directionality algorithm were previously described in Lassers et al. [19]. To briefly summarize the analysis pipeline, raw tunnel data sampled at 25 kHz were filtered through Wave_Clus [20], and spike detection and clustering were computed from 5 to 50 S.D. noise and 50.1 to 500 S.D. to ensure the counting and clustering of large axonal spiking. A refractory period of 1.5 ms was specified. Any spike shapes differing by less than three standard deviations from the mean spike shape were included in a single cluster. The large, tolerated deviation was chosen to accommodate different axon-electrode coupling for a single axon on each of the two electrodes. Clustering was used to discard complex spikes from multiple axons in a microtunnel that produce overlapping spike waveforms. Similar to previous analyses of these tunnel devices that demonstrated that ~63% of tunnels contained only one axon [21], single axons were identified by their uniform conduction velocities or spike timing delays. These timing delays were used to generate a normalized matching indexing (NMI) algorithm which was computed for every tunnel using the timing comparison made from the two electrodes spanning each tunnel (Eq. (1)). All tunnels were the same length of 400 µm except for the EC-CA3 diagonal tunnels that ranged from 400–600 µm. To rigorously determine which tunnels have single axons so that directionality could be determined, NMI was calculated with the equation:

$$NMI = \frac{\# \ paired \ spikes}{max(total \ \#spikes \ per \ spike \ train)}$$

(1)

Tunnels with NMI > 0.2 (20% of spikes matched between two clusters) were considered valid. This threshold was sufficient for eliminating spurious spike pair correlations during high spike rates.

A histogram of conduction times was generated with thresholding and peak prominence values provided by the MATLAB findpeaks function. Valid delay times were between 0.2 ms and 1 ms if there was a peak at sufficiently fast conduction times. Feed-forward axons were identified by positive delay times and feed-back axons by negative conduction times, the basis for our determination of the feed-forward and feedback directionality of axonal communication. In contrast, raw

 

data from the wells were filtered through Wave_Clus and spikes were detected at the threshold of ±3.5 S.D. noise. This lower threshold than for axons better accommodated the lower signal to noise ratio in the wells. Spike clustering was not needed since single neurons were detected in >90% of cases.

### Spike dynamics and probability distributions

Vakilna et al. [8] showed that the distribution of inter-spike intervals (ISI) and inter-burst intervals (IBI) follow log–log distributions and were visualized as normalized complementary cumulative probability distributions (CCDs) with logarithmically spaced bins [22]. A log-transformed linear model of the slope, m, and intercept c was used to fit the CCD after log transformation.

$$log_{10}(P) \; = \; m \times log_{10}(t) + \; c$$

(2)

$$(from \; P = t^m c)$$

A grid search of the local maximum for $Pearson's \; R^2$ was used to find the best fits with time limits varied up to 50% with a step size of 5%. A single fit was found for all ISI CCDs over the probability range from 1 to 0.1. The four-tunnel ISI CCDS intervals were different for each tunnel, for both feed-forward and feedback axons. For the five-tunnel design, intervals were from 0.01 to 0.2 s except for CA1-EC feed-forward and feedback axons which were fit for intervals between 0.01 to 0.11 s and 0.01 to 0.09 s respectively, to account for non-linearities in the distribution. Two linear fits were calculated for all inter-burst interval (IBI) CCDs piecewise to account for the "up states" and "down states," referring to fast and slow bursting, respectively [8]. The minimum time for the up states was used as the maximum time for the down states. We compared the distribution of ISI to a log-normal distribution instead of a log-log (S1 Fig) to show that the distributions were not log-normal. The burstiness of a neuronal unit was defined as the percentage of tagged spikes that appear in bursts, at least four spikes, each separated by less than 50 ms. Distributions of axonal and somal burstiness were plotted normalized by the number of arrays in each network design and statistical tests were performed to assess changes in burstiness between designs.

### Statistics

Data were analyzed with custom MATLAB 2024b scripts. Slopes were compared for significant statistical differences at alpha = 0.05 using analysis of covariance (ANCOVA) followed Tukey's honest significant difference (HSD) test. The burstiness of the data was compared using the nonparametric Kruskal-Wallis test since the distributions of burstiness had significant deviations from the normal distribution to determine if the data came from different distributions at $p < 0.05$. Data were combined and analyzed for nine separately plated networks for the four-tunnel model and six separately plated networks for the five-tunnel model.

### Ethics statement

The animal study was approved by UC Irvine Institutional Animal Care and Use Committee. The study was conducted in accordance with the local legislation and institutional requirements.

## Results

### Different subregional and axonal spiking and bursting dynamics between four-tunnel and five-tunnel self-wired hippocampus model

To explore the impact of hippocampal architecture on spike dynamics, we designed a four-compartment device with micro-fluidic tunnels connecting sub-regions that promoted axonal self-wiring and prevented dendritic infiltration (Fig 1). Each of

the four compartments contained a sub-region of the trisynaptic loop (EC, DG, CA3, and CA1) and tunnels between each sub-region for their communication. The traditional method of studying hippocampal information encoding interprets behavior from electrical recording of only one hippocampal subregion, most commonly the CA1, which introduces biased attention to an individual part of the hippocampus. This method often ignores the differences in dynamics of information encoding in each sub-region of the hippocampus or in control of the axonal information that gets transmitted from one region to the next. Efforts to find mono-synaptic connections between somata in two subregions by paired two site intracellular recording in slices are faced with low probabilities, such as 1.3% feedback from CA3 to the dentate [23] or the need to identify 5–10 CA3 pyramidal cells to sum EPSP's within 50 ms to excite a single CA1 target [24]. However, this approach is often used because of the difficulty in accessing all regions of the hippocampus simultaneously and the impossibility of isolation of recordings from single axons during in vivo recordings. Superimposing the four-compartment device over a multi-electrode array allowed recording of spontaneous activity. Our four-compartment design enabled monitoring of subregional and between region activity in single axons. As previously demonstrated for axons communicating between two compartments of DG and CA3 [21], these micron-sized tunnels mostly contained only one or two axons each.

## Dynamics of feed-forward and feedback axons reveal faster spiking in five-tunnel vs. four-tunnel architecture

The spatiotemporal organization of spikes may provide insight to the function and behavior of the hippocampal network. Information relayed between sub-regions of the hippocampus was analyzed through the neuronal spiking in communicating axons. Dual electrodes were used to record spike timing differences and determine the direction of spike propagation [8]. We define feed-forward as EC > DG > CA3 > CA1, back to EC in the trisynaptic loop and the reverse, if detected, as feedback timing of action potential propagation. Feed-forward activity was likely excitatory and feedback likely inhibitory. The one exception to this is that the axons that synapse with the CA1 from the EC are likely excitatory. When axons in similar devices of CA3 connected to DG were stained for the GABA synthesis enzyme, glutamic acid decarboxylase (GAD), strong staining indicated inhibitory axon transmission [25]. The role of EC-CA1 is evaluated as feed-forward because it is most likely an excitatory pathway [26]. As previously found in Vakilna et al. [8], spontaneous interspike intervals (ISI) followed log-log distributions in the axons between subregions (Fig 2, S1 Table 1 statistics). A linear model was fit to the data by Eq. (2) to describe the probability (P) of a spike firing at time (t) after a previous spike and m as the slope of the fit. Thus, graphs with steeper slopes represent shorter times which represent faster firing rates. Fig 2 shows the spike timing in the four-tunnel architecture compared to the five-tunnel architecture as cumulative probability distributions of inter-spike intervals (ISI), both in feed-forward (A–H) and feedback (I–N) axons by subregion. All distributions were well-fit by a single exponential slope with $R^2 > 0.99$, suggesting a common mechanism of spike timing. All slopes were significantly different by ANCOVA followed by Tukey-HSD (S1 Table). Fig 2F shows that most of the four-tunnel feed-forward slopes were significantly different (partial eta$^2$ effect size 0.96), as were most of the five-tunnel slopes (Fig 2G, effect size 0.91). Fig 2H compares the slopes of the four to the five-tunnel architectures where feed-forward spike rates were 17% faster for the four-tunnel vs. the five-tunnel configuration at the first EC-DG stage of transmission. In all other connections, slopes in the five-tunnel design were 30–87% faster than the four-tunnel axon architectures with a large effect size of 0.16. Compared to the four-tunnel configuration, this means higher overall spike rates in the five-tunnel enabled by the additional EC-CA3 axonal communication.

We conducted similar analyses of the subregions that either send or receive the axons monitored in the tunnels (Fig 3, S2 Table statistics). For four-tunnel networks, comparison of Fig 3A–D shows the slowest spiking in CA3 (Fig 3E, effect size 0.93). Similar comparisons for the five-tunnel networks show the fastest spiking in EC compared to the other sub-regions (Fig 3F, effect size 0.89). Comparison of the two tunnel types in Fig 3G, shows only a 4% steeper slope in EC for the five-connect networks, while all the others produced an 11–80% steeper slope for the four-connect networks over those of the five-connect (effect size 0.02 for interaction; 0.32 for 4 v 5 model). Next, we explored the difference in grouping of spikes into bursts and its impact on network communication.

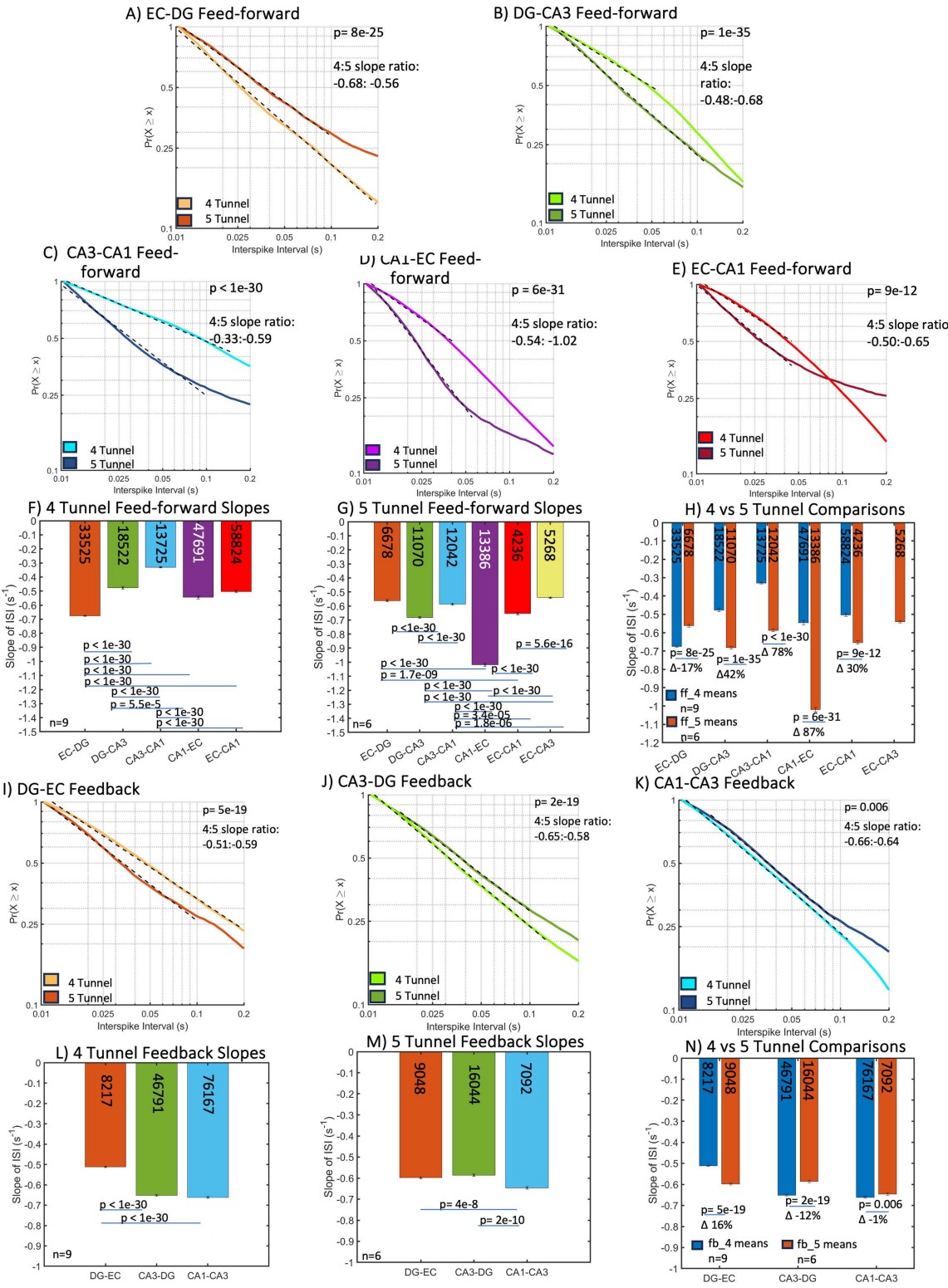

**Fig 2. Axonal cumulative probability distributions of inter-spike intervals (ISI) and slopes in the four-tunnel and five-tunnel architectures.** (A–H) feed-forward (FF); (I–N) feedback (FB) axons. (A) EC-DG four-tunnel axons have 17% faster spiking than five-tunnel FF ISI slope. (B) DG-CA3 five-tunnel axons have 42% faster spiking than four-tunnel FF ISI slope. (C) CA3-CA1 five-tunnel axons have 78% faster spiking than four-tunnel FF ISI

slope. (D) CA1-EC five-tunnel axons have 87% faster spiking than four-tunnel FF ISI slope. (E) EC-CA1 five-tunnel has 30% faster spiking than four-tunnel FF ISI slope. (F) All four-tunnel feed-forward axonal slopes are significantly different. (G) Most of the five-tunnel slopes are significantly different with CA1-EC the fastest. (H) All slopes are significantly different between four-tunnel and five-tunnel feed-forward axons. (I–N) feedback (FB). (I) DG-EC five-tunnel axons have 16% faster spiking than four-tunnel FB ISI slope. (J) CA3-DG four-tunnel axons have 12% faster spiking than five-tunnel FB ISI slope. (K) CA1-CA3 four-tunnel axons have 1% faster spiking than five-tunnel FB ISI slope. (L) Most of the four-tunnel feedback slopes are significantly different except CA3-DG and CA1–CA3. (M) Most of the five-tunnel CA1-CA3 feedback slopes are significantly different from other tunnels. (N) All feedback slopes are significantly different between four and five-tunnel wiring, but differences are small. P-values calculated using ANCOVA followed by Tukey-HSD. Numbers within bars represent total ISIs from 9 and 6 arrays, respectively. (I) N shows the spike timing in feedback axons. In L, both CA3 feedback into DG and CA1 into CA3 spike rates were higher than DG to EC feedback in the four-tunnel architecture (effect size 0.92). Feed-back spike rates within five-tunnel axons were increased by 16% for DG back to EC compared to the other feedback routes (M, effect size 0.41). In N, we compare feedback rates for 4 vs. five-tunnel architectures where spike rates differed by only -12 to +16% (effect size 0.00), much less than the -17 to 87% differences in the feed-forward configuration (H, effect size 0.16). S2 Fig shows the same data rearranged to compare different inter-subregion tunnel axons.

## Slower bursting overall in five-tunnel versus four-tunnel architecture

Spikes are organized into bursts or packets of information for both local and global computation and routing [27–29]. These bursts of spikes are easily recorded *in vivo* because of their intrinsic function in raising somal calcium concentration [30]. Each sub-region of the hippocampus is responsible for a different set of computations that give rise to learning and memory [2]. The spatiotemporal bursting dynamics in sub-regions and axons is key for insight into each sub-region's function in routing information for processing. Previously, we found spontaneous interburst intervals (IBI) followed a two-part log–log distribution suggesting faster up and slower down states that vary sub-regionally and between regions [8]. Like ISI, a decrease in slope indicates a slower frequency of bursting.

Feed-forward fast bursting and slow bursting (Fig 4, S3 Table statistics) generally decreased in five-tunnel compared to four-tunnel axon architecture. Bursts were plotted over a timescale of two orders of magnitude representing probability vs. times at which a bursting event occurred. Here, we focused on fast bursts as they represent most events. In the four-tunnel axons, bursting was slowest in the EC-DG (Fig 4A, effect size 0.90) and CA3-CA axons (Fig 4C, effect size 0.00), as compared in Fig 4E (effect size 0.90). Among four-tunnel axons (Fig 4F), feed-forward burst rates were fastest in CA1-EC and high in DG-CA3 and EC-CA1 compared to EC-DG and CA3-CA1 (effect size 0.98). Among five-tunnel axons, these regions were joined in slower bursting by EC-CA1 and EC-CA3 (Fig 4G, effect size 0.90). In comparison to the four-tunnel architecture (Fig 4H, effect size 0.10), feed-forward five-tunnel networks burst rates were 30–76% slower with EC-CA1 showing the largest difference. On average, the four-tunnel arrays had 943 bursts per recording network-wide with a standard deviation of 695 and the five-tunnel arrays had 142 bursts per recording with a standard deviation of 58 suggesting that the five-tunnel arrays have less variance in their burst rates compared to the four-tunnel arrays. Fig 4I, J, K compare the slopes of the feedback bursting. In the four-tunnel networks (Fig 4L, effect size 0.90), feedback bursting was highest in the CA1 to CA3 networks. In the five-tunnel networks (Fig 4M, effect size 0.92), bursting was 52–56% lower than in the four-tunnel networks (Fig 4L). In Fig 4N (effect size 0.06 for the interaction; 0.24 for the 4 v 5 model) the feedback axons for the four-tunnel arrays had an average of 2169 bursts per recording with a standard deviation of 1260 (excluding the outlier DG-EC connection) and the five-tunnel arrays had an average of 219 bursts per recording with a standard deviation of 73, again showing greater control of bursting activity with less variance in the five-tunnel arrays.

Sub-regions showed an overall decrease in bursting in the five-tunnel compared to the four-tunnel networks (Fig 5A, B, C, D, S4 Table statistics, effect sizes 0.96–0.98). Subregions EC, DG and CA1 neurons burst at 80–100% higher rates than CA3 in the four-tunnel architectures (Fig 5E, effect size 0.81) and burst at 42–76% slower rates (longer interburst intervals) in the five-tunnel architecture (Fig 5F effect size 0.50) as compared in Fig 5G, effect size 0.08 for the interaction, 0.31 for the 4 v 5 model) with CA1 showing the largest difference compared to their four-tunnel counterparts.

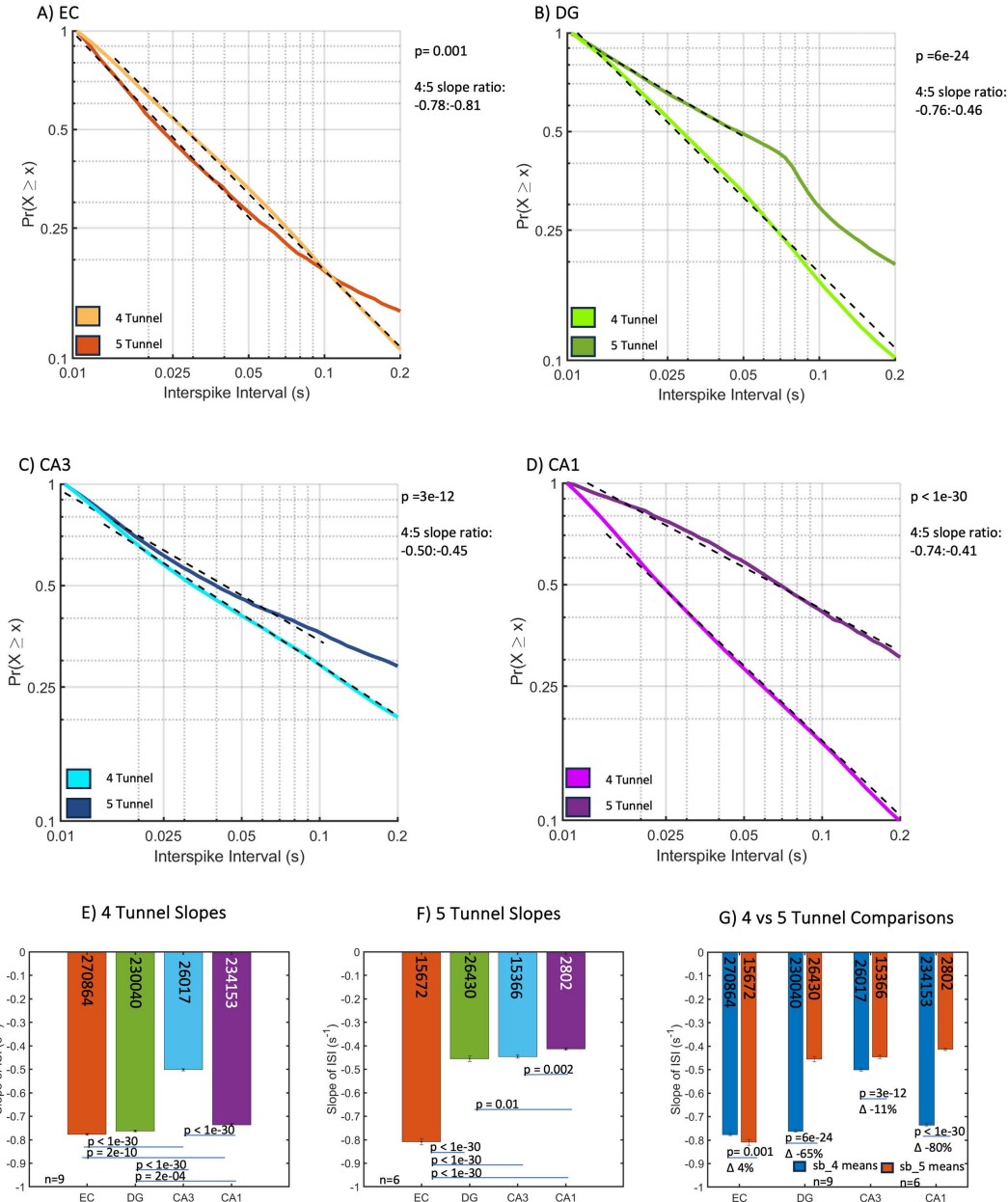

**Fig 3. Subregional cumulative probability distributions of inter-spike interval (ISI) slopes in the four-tunnel (*n* = 9 arrays) and five-tunnel architectures (*n* = 6 arrays).** The four-tunnel axon architecture promotes faster subregional spiking than the five-tunnel axon architectures in all subregions. (A–G) subregional (A) EC five-tunnel has 57% slower spiking than four-tunnel ISI slope. (B) DG five-tunnel has 37% slower spiking than four-tunnel ISI slope. (C) CA3 five-tunnel has 45% slower spiking than four-tunnel ISI slope. (D) CA1 five-tunnel has 54% slower spiking than four-tunnel ISI slope. (E) Most four-tunnel subregion slopes were significantly different from each other except EC and CA1. (F) Five-tunnel subregion slopes were similar except for differences from DG, and DG vs CA1. (G) All subregions are significantly different between four-tunnel and five-tunnel. P-values calculated using ANCOVA followed by Tukey-HSD. Numbers within bars represent total ISIs from 9 and 6 arrays, respectively.

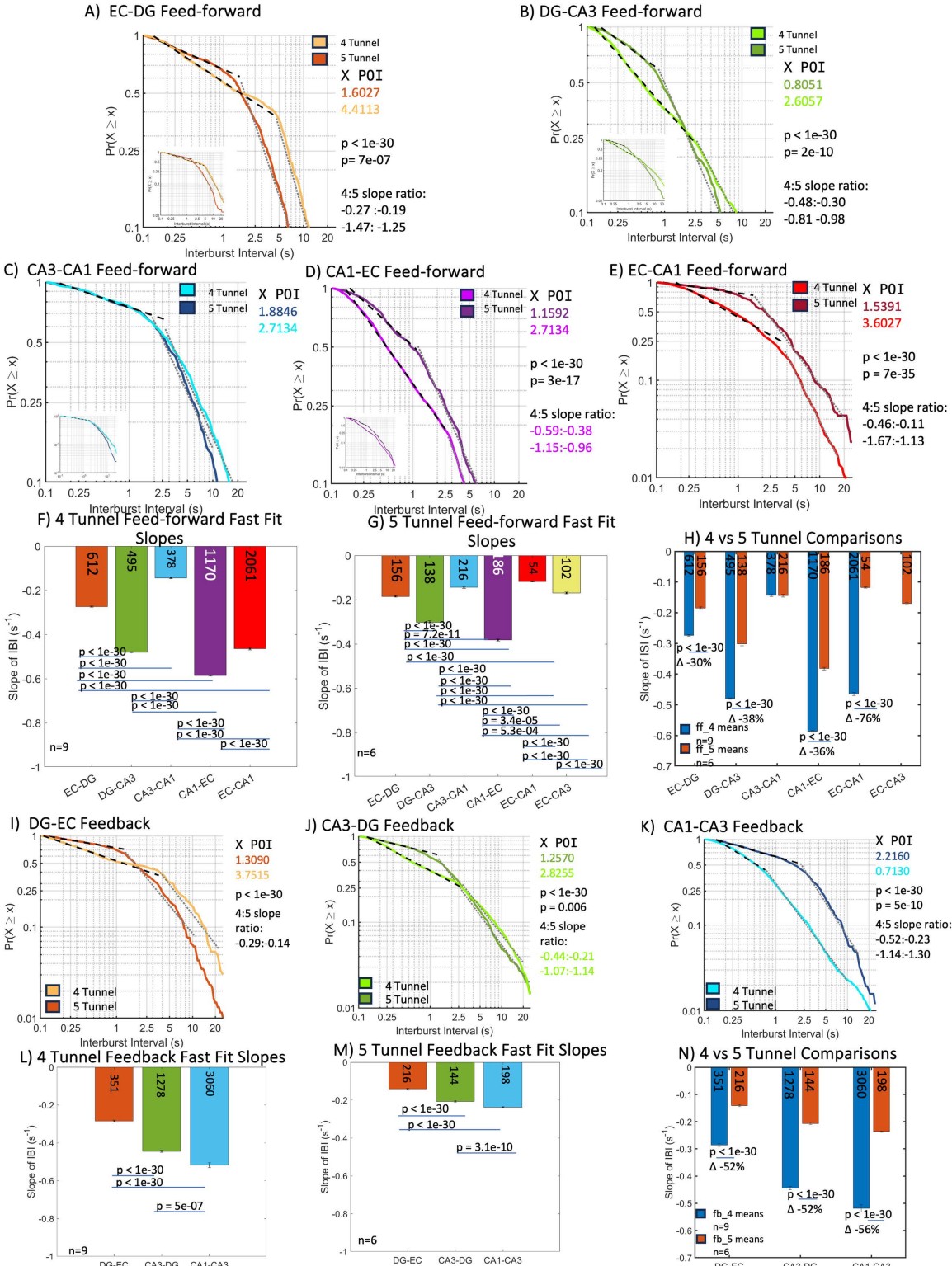

**Fig 4. Smaller differences in interburst interval (IBI) slopes in inter-regional axon distributions in the four-tunnel (*n* = 9 arrays) and five-tunnel architecture (*n* = 6 arrays).** The five-tunnel feed-forward axon architecture shows a switch from short interburst intervals to long interburst intervals earlier than the four-tunnel axon architectures. (A–H) feed-forward (FF) (A) EC-DG fast fit four-tunnel has 30% faster bursting than five-tunnel FF IBI slope.

(B) DG-CA3 fast fit four-tunnel has 38% faster bursting than five-tunnel FF IBI slope. (C) CA3-CA1 fast fit had no significant difference between four-tunnel and five-tunnel architectures. (D) CA1-EC fast fit four-tunnel has 36% faster bursting than five-tunnel FF IBI slope. (E) EC-CA1 fast fit four-tunnel has 76% faster bursting than five-tunnel FF IBI slope. (F) Four-tunnel feed-forward slopes are significantly different except DG-CA3 and EC-CA1. (G) Most of the five-tunnel slopes are significantly different. (H) All slopes are significantly different between four-tunnel and five-tunnel except CA3-CA1. (I–N) feedback (FB) (I) DG-EC fast fit four-tunnel has 52% faster bursting than five-tunnel FB IBI slope. (J) CA3-DG fast fit four-tunnel has 52% faster bursting than five-tunnel FB IBI slope. (K) CA1-CA3 fast fit four-tunnel has 56% faster bursting than five-tunnel FB IBI slope. (L) All four-tunnel feedback slopes are significantly different. (M) Five-tunnel feedback slopes are significantly different except CA1-CA3 and CA3-EC. (N) All slopes are significantly different between four and five-tunnel. P-values calculated using ANCOVA followed by Tukey-HSD. POI (position of intersect) is the intersection of the fast fit and slow fits. Numbers within bars represent total IBIs from 9 and 6 arrays, respectively.

### CA3-CA1 and CA1-EC feed-forward burstiness increases in the presence of the EC-CA3 perforant pathway

We measured the burstiness in axons and soma in our networks as the fraction of spikes in bursts for each subregion and interregion, a distribution of burstiness in axons and soma were gathered and the fraction of units with spikes in bursts above 50% was calculated. Under the paradigm of the packet switching brain, bursts represent parts of an encoded message. This can be interpreted as active processing [28,29]. With the addition of the five-tunnel architecture, we observed an increase in feed-forward burstiness in the CA3-CA1 and CA1-EC pathways by comparison through a nonparametric Kruskal–Wallis, a test for differences between two distributions (Fig 6A). *In vivo*, the perforant pathway from the EC to CA3 is responsible for encoding ques that can reconstruct memories, those memories are then organized in the CA1 and back propagated to the EC and to the cortex for higher level processing. With the addition of the EC-CA3 pathway, we can now see encoded messages being sent from the CA3 to the CA1 and the CA1 sending those messages to the EC. Differences in burstiness in the feedback pathways were not significant (Fig 6B). The burstiness of the DG and CA1 subregions decreased in the five-tunnel design (Fig 6C).

## Discussion

### More inter-regional feed-forward spiking but less inter-regional bursting with feed-forward EC-CA3 inputs versus four-tunnel architecture without direct EC to CA3 input

Using a novel four-compartment device, we were able to reconstruct a hippocampal network comprised of the entorhinal cortex, dentate gyrus, CA3, and CA1 subregions in a loop with axonal connections isolated in microfluidic tunnels. Each tunnel and subregion were equipped with electrodes that enabled the monitoring of the direction of axonal propagation. Previously, connectivity of the trisynaptic circuit was inferred through changes in somal activity of other sub-regions. This novel system used self-wiring that allowed differences in axonal spike dynamics to be measured by sub-region. All spiking dynamics followed linear log-log distributions, displaying the canonical non-Gaussian distributions inherent to computations in neural systems [7]. As previously reported by [8], interburst intervals required two slopes to fit a linear log-log model, suggesting up- and down-states of network computation.

Between the four-tunnel and five-tunnel model, the five-tunnel architecture showed an increase in feed-forward firing, but bursting decreased by up to 76% in both soma and axons in the feed-forward and feedback directions. This suggests tighter control of spiking and more precise communication between subregions. Indeed, the additional EC-CA3 layer skipping connection added 30% more axonal connectivity. However, we expect that adding more tunnels alone is unlikely to result in the large changes in spiking and burst behavior that we see due to asymptotic behavior in comparisons of 10, 15 and 51 tunnels [31]. Information transfer in the five-tunnel appears more organized than in the four-tunnel network, as we have seen with these same networks after stimulation [19].

### Function of the EC-CA3 perforant path

The EC-CA3 connection in the perforant pathway plays a crucial role in modulating hippocampal dynamics and functionality. Activation of the CA3 region is largely dependent on feed-forward excitatory inputs from direct perforant path fibers

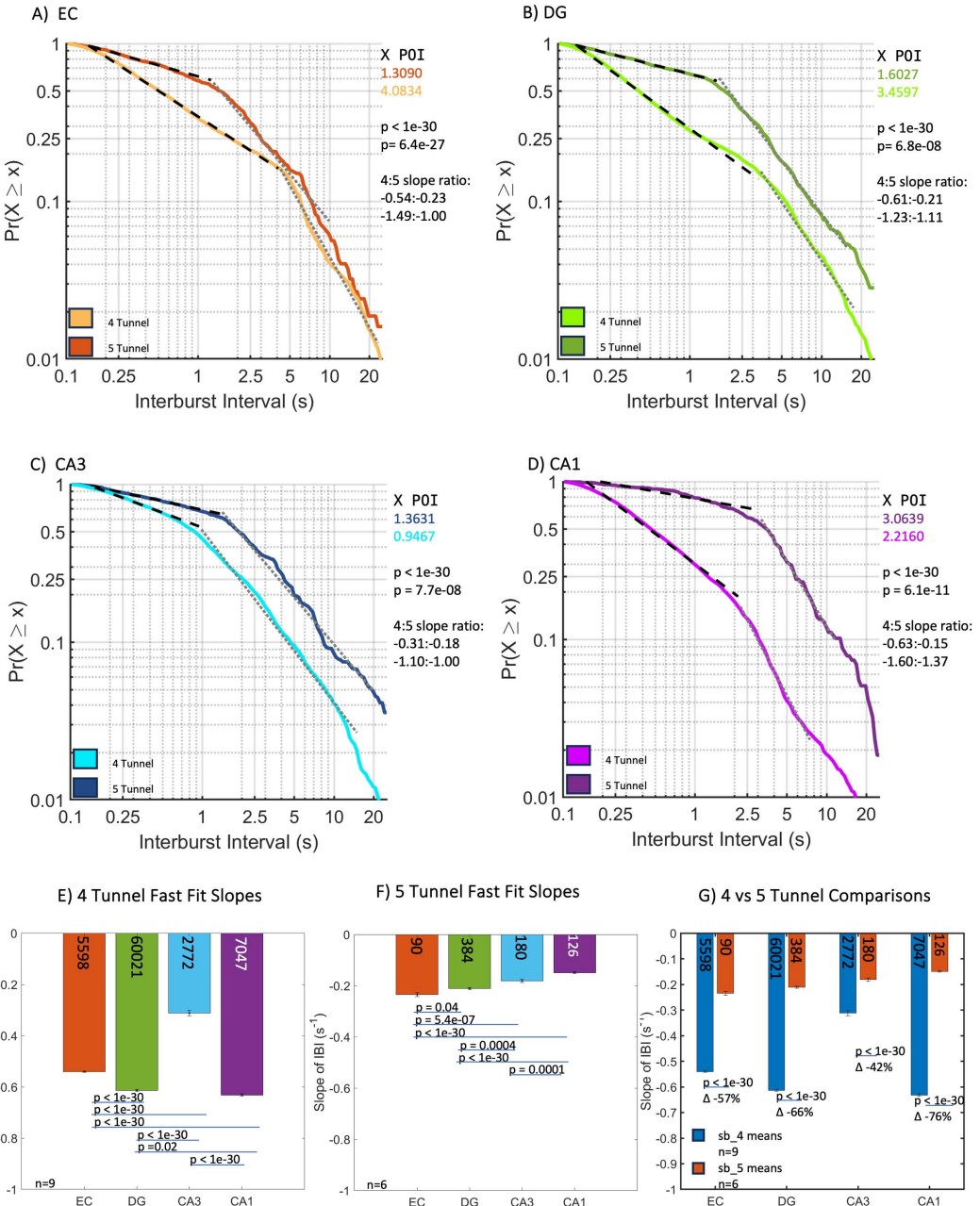

**Fig 5. Subregional interburst interval (IBI) distributions of slopes in the five-tunnel architecture (_n_ = 6 arrays) burst slower than four-tunnel architecture (_n_ = 9 arrays).** (A–G) subregional (A) EC fast fit five-tunnel has 70% slower bursting than four-tunnel IBI slope. (B) DG fast fit five-tunnel has 66% slower bursting than four-tunnel IBI slope. (C) CA3 fast fit five-tunnel has 40% slower bursting than four-tunnel IBI slope. (D) CA1 fast fit five-tunnel has 66% slower bursting than four-tunnel IBI slope. (E) All four-tunnel subregion slopes were significantly different from each other except DG vs CA1. (F) All five-tunnel subregion slopes were significantly different from each other except EC vs DG and DG vs CA1. (G) All subregions are significantly different between four-tunnel and five-tunnel. P-values calculated using ANCOVA followed by Tukey-HSD. POI is the intersection of the fast fits and slow fits. Numbers within bars represent total IBIs from 9 and 6 arrays, respectively.

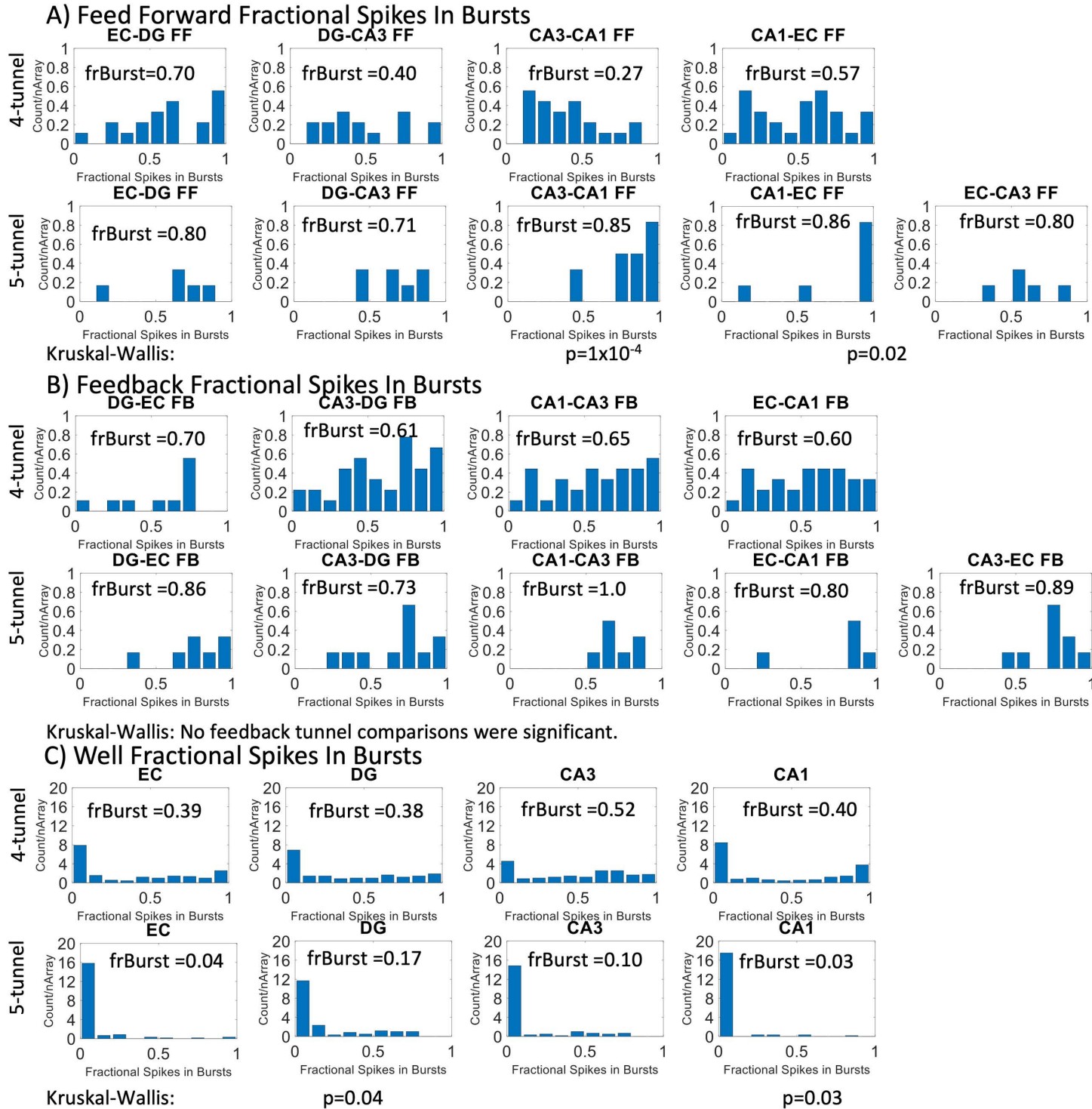

**Fig 6. Fraction of axons with spikes appearing in bursts (frBurst) was higher for CA3-CA1 and CA1-EC feed-forward axons in five-tunnel arrays compared to four-tunnel arrays, normalized by the number of arrays for each design.** A Kruskal–Wallis test was performed to assess significant differences in burstiness. (A) Burstiness in four-tunnel and five-tunnel architecture for feed-forward axons. The fraction of bursts in CA3-CA1 and the CA1-EC feed-forward tunnels in the five-tunnel arrays were significantly higher than the four-tunnel arrays. (B) Burstiness for feedback axons. There are no significant differences between the two array designs. **(C) Burstiness** for subregional spikes. DG, and CA1 were significantly lower in the five-tunnel architecture according to Kruskal–Wallis test.

[32]. Our networks show a large amount of feed-forward activity between all subregions, following the canonical representation of the trisynaptic loop. The five-tunnel architecture showed 30–87% greater interregional feed-forward spiking with the CA1-EC connection showing the greatest increase. Lesions in the CA3 region have been shown to impair spatial recall without significantly modifying CA1 activity, while lesions in the EC input led to mild impairments despite reducing CA1 spatial selectivity [33]. Therefore, the EC-CA3 seems to enable spatial navigation. Feedback spike rates differed to a much smaller extent from 12 to 16% compared to the feed-forward configuration. Compared by subregions, network differences produced an 11–80% advantage in axonal spiking for the four-tunnel architecture over the five-tunnel architecture. Degradation of these fibers is correlated with increased CA3 hyperactivity [5]. It may be that inputs from the EC help regulate CA3 in a more precise fashion. This is supported by the 11% gain in feed-forward activity of the CA3 subregion in the five-tunnel configuration. To hypothesize about these effects on bursting in the context of the larger scale of episodic memory, the increase in CA1-EC bursting activity in the presence of the EC-CA3 pathway may be indicative of dynamics that contribute to routing time-ordered episodes back to the EC for back projection to the cortical layers. This is in contrast to DG mossy fiber inputs to the CA3 causing long term potentiation for learning which may only need to be isolated to the CA3. So in the absence of a layer skip, information from the EC was diluted in the DG. Because we did not stimulate the cultures in this study, the DG may not have needed to encode new patterns and excite the CA3 network. However, skipping the DG, the EC to CA3 can be a continuously excitatory process that evokes activity for recall instead of long-term potentiation. This could be one reason why more spikes were found in bursts, or more structured bursting, in the five-tunnel than the four-tunnel.

Balancing and modulating excitatory/inhibitory (E/I) activity is crucial for efficient memory encoding [34]. Spontaneous firing patterns of neurons help understand network states and stimulus representation. Achieving global E/I balance requires sparsely connected neurons with strong inhibitory connections to prevent excessive excitation [34]. Pyramidal neurons maintain optimal firing rates for information processing, while inhibitory interneurons respond over a wider range, aiding in local homeostasis and sub-regional coordination range [35]. The minimal differences we found in feedback spiking dynamics suggest that excitatory and inhibitory signals are more balanced in the five-tunnel architecture.

In contrast, the five-tunnel architecture produced burst rates 30–76% slower than the four-tunnel architecture, both interregional and by subregion, with CA1 connections showing the greatest difference in burst rates. The brain's circuits do not produce the same response to identical input each time. This variability reflects brain network plasticity. Information in the brain, especially between hippocampal sub-regions, is likely transmitted in bursts of action potentials. These bursts can be compared to packets in Internet communication, where information is sent in parts and then reassembled or interpreted at a specific location, enhancing message reliability [27]. In cortical circuits, bursting packets are essential for information transmission. The number and timing of spikes within these bursts convey details about stimulus types, as was previously identified between the DG and CA3 [36]. Control experiments by Brewer et al. [25] revealed unique spike dynamics between DG-CA3, distinct from DG-DG and CA3-CA3, indicating that specific anatomical structures result in distinct coding patterns.

The addition of the EC-CA3 connection decreased bursting IBI in the five-tunnel architecture and specifically in CA1 axons. Burst activation for the EC-CA1 pathway is known to enhance long-term potentiation [33]. However, increased spiking in this pathway suggests spike timing may be better controlled with more spikes outside bursts. Despite minor changes in spike dynamics in DG-EC feedback, the higher number of spikes may indicate the development of more inhibitory neurons in DG to more finely control network dynamics. The overall differences in spiking and bursting rates between four and five-tunnel architectures suggest there may be a balancing mechanism related to the spatial-temporal organization of spikes.

## The packet paradigm of information transfer in networks

Based on bursting spikes associated with behavior in the rat brain, bursts are proposed to operate as packets of information transfer [29]. The theoretical framework of Graham and Rockmore [27] proposes that certain brain networks have the

ability to "packet switch" analogous to packets of data transferred over the Internet. Packets are pieces of a whole message where each piece is sent along the most efficient route and recombined at the target. When a packet is received, the target sends a message back confirming its receipt. This reduces the delay of the message being received and increases the bandwidth of the communication network. In this metaphor, packets are analogous to bursting activity in biological neural networks and feedback network activity could function like a receipt of information. The most interesting difference we have seen in comparing our addition of an EC-CA3 connection in the network design is the increase in burstiness of the CA3-CA1 feed-forward connection with more spikes and bursts per array than in the four-tunnel design. This suggests that the EC-CA3 connection enhances the sharpness or strength of information routing from the CA3 to the CA1 by tightening the specificity of spiking within bursts. In vivo recordings of CA1 in neonatal rats, support these findings with a high fraction of spikes in bursts over the same range of 0.2–20 s interburst intervals [37].

## Neuromorphic computing applications

Neuromorphic computing is a bio-inspired computing paradigm that leverages event-based computing and neuronal architectures to solve brain-like problems including deep learning and continuous learning at low power consumption. Recent advances in artificial neural network architectures have started to explore layer skipping as a way to avoid optimization problems and generalize the performance of neural networks with less information loss [38]. The information skipped past the previous layer is re-integrated after the hidden layer operations. However, the reasons why layer skipping creates the ability to train very deep networks are unclear. Layer skipping is intrinsic to the hippocampus through the EC-CA3 pathway studied here and may act as a way to prime downstream neurons for input from the DG to the CA3. Our laboratory is beginning investigations into hippocampus-inspired neural networks as a way to create generalizable auto-associative *in silico* neural networks. These results suggest that more collaboration between the neuroscience and the machine learning communities for new computing architectures could accelerate advances in artificial intelligence.

## Limitations

The *in vitro* model network developed to study spike dynamics may not accurately reflect the complexity of *in vivo* conditions for several reasons. The culture is two-dimensional, simplifying the complex 3D axonal organization seen *in vivo*, which enables more sophisticated signal integration. Although some functional connectivity in the hippocampus occurs within a specific thickness, the *in vitro* model lacks the laminated arrangement of neurons seen in the hippocampus, affecting axonal innervation patterns in specific dendritic layers. The slower conduction velocity of our model unmyelinated axons are unlike the myelinated perforant path in vivo. 3D cultures, which more closely resemble *in vivo* systems, show greater response to stimuli but have not yet achieved isolated axon cultures as with the current model. The model also faces limitations in electrophysiological monitoring and the number of neurons and axon tunnels that can be analyzed at once, affecting the thoroughness of functional connection analysis. Anatomical differences from *in vivo* systems include the lack of inputs from other brain regions, including modulatory cholinergic and dopaminergic inputs, which may limit our understanding of network behavior under input conditions of freely behaving animals. Also, the observation of feedback activity from CA3 to EC (not shown) indicates that some unphysiological connections are also possible when evidence for this connection is lacking in vivo [39]. Future work includes improving the model's 3D culture and electrophysiological monitoring capabilities and using optogenetics for defined multicell patterned stimulation to more closely mimic *in vivo* conditions.

## Conclusion

Our study revealed significant differences between the four-tunnel and five-tunnel hippocampal models, providing new insight into the role of the EC-CA3 perforant pathway's computational impact. The addition of the perforant path-like EC-CA3 connections in the five-tunnel architecture resulted in shorter inter-spike interval distributions balanced by longer

interburst intervals. We observed distinct spiking patterns in each of the four subregions and in their communicating axons, indicating functional differences in information processing between subregions and between the four-tunnel and five-tunnel architecture. The inclusion of the EC-CA3 pathway in the five-tunnel design led to more structured communication between subregions with a different E/I balance, as evidenced by the altered spike dynamics. Furthermore, we noted increased burstiness in the CA3-CA1 and CA1-EC feed-forward pathways in the five-tunnel design, suggesting enhanced information packet coding. These findings highlight the role of specific neural pathways in shaping network dynamics and may inform future computational models and understanding of hippocampal function in learning and memory processes.

## Supporting information

**S1 Fig. Semi-log comparisons of axonal cumulative probability distributions of inter-spike intervals (ISI) and slopes in the four-tunnel ($n$ = 9 arrays) and five-tunnel architecture ($n$ = 6 arrays) for each subregion.** Both four-tunnel and five-tunnel axon architectures show non-Gaussian semi-log distributions of ISIs. (A–D) feed-forward (FF). (E–H) feedback (FB).
(TIF)

**S2 Fig. Subregional comparisons of axonal cumulative probability distributions of inter-spike intervals (ISI) and slopes in the four-tunnel ($n$ = 9 arrays) and five-tunnel architecture ($n$ = 6 arrays) compared by subregions.** (A and B) axonal feed-forward (FF). (C and D) axonal feedback (FB). (E and F) subregional.
(TIF)

## Author contributions

**Conceptualization:** Samuel B. Lassers, Gregory Brewer.

**Data curation:** Ruiyi Chen, Gregory Brewer.

**Formal analysis:** Samuel B. Lassers, Shazfa S. Khatri.

**Funding acquisition:** Gregory Brewer.

**Investigation:** Samuel B. Lassers, Shazfa S. Khatri, Gregory Brewer.

**Methodology:** Gregory Brewer.

**Project administration:** William C. Tang, Gregory Brewer.

**Resources:** Gregory Brewer.

**Software:** Samuel B. Lassers, Shazfa S. Khatri, Yash S. Vakilna.

**Supervision:** William C. Tang, Gregory Brewer.

**Validation:** Samuel B. Lassers, Shazfa S. Khatri, Gregory Brewer.

**Visualization:** Samuel B. Lassers, Shazfa S. Khatri.

**Writing – original draft:** Samuel B. Lassers, Shazfa S. Khatri.

**Writing – review & editing:** Samuel B. Lassers, Shazfa S. Khatri, Ruiyi Chen, Yash S. Vakilna, William C. Tang, Gregory Brewer.

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
