## [Decision Letter · Decision Letter 0]

PONE-D-24-49403Impact of the entorhinal feed-forward connection to the CA3 on hippocampal codingPLOS ONE

Dear Dr. Brewer,

Thank you for submitting your manuscript to PLOS ONE. After careful consideration, we feel that it has merit but does not fully meet PLOS ONE’s publication criteria as it currently stands. Therefore, we invite you to submit a revised version of the manuscript that addresses the points raised during the review process.

We look forward to receiving your revised manuscript.

Kind regards,

Lei An

Academic Editor

PLOS ONE

3. Thank you for stating in your Funding Statement: [The author(s) declare financial support was received for the research, authorship, and/or publication of this article. This study was supported in part by funds from the UC Irvine Foundation. The Foundation had no decision making capacity for the design or manuscript].

Please provide an amended statement that declares *all* the funding or sources of support (whether external or internal to your organization) received during this study, as detailed online in our guide for authors at http://journals.plos.org/plosone/s/submit-now . Please also include the statement “There was no additional external funding received for this study.” in your updated Funding Statement.

4. Thank you for stating the following financial disclosure: [The author(s) declare financial support was received for the research, authorship, and/or publication of this article. This study was supported in part by funds from the UC Irvine Foundation. The Foundation had no decision making capacity for the design or manuscript].

5. Thank you for uploading your study's underlying data set. Unfortunately, the repository you have noted in your Data Availability statement does not qualify as an acceptable data repository according to PLOS's standards.

6. We notice that your supplementary figures are included in the manuscript file. Please remove them and upload them with the file type 'Supporting Information'. Please ensure that each Supporting Information file has a legend listed in the manuscript after the references list.

Reviewers' comments:

Reviewer's Responses to Questions

**Comments to the Author**

1. Is the manuscript technically sound, and do the data support the conclusions?

Reviewer #1: Yes

Reviewer #2: No

Reviewer #3: Yes

Reviewer #4: Yes

2. Has the statistical analysis been performed appropriately and rigorously? 

Reviewer #1: Yes

Reviewer #2: Yes

Reviewer #3: Yes

Reviewer #4: No

3. Have the authors made all data underlying the findings in their manuscript fully available?

Reviewer #1: Yes

Reviewer #2: Yes

Reviewer #3: Yes

Reviewer #4: Yes

4. Is the manuscript presented in an intelligible fashion and written in standard English?

Reviewer #1: Yes

Reviewer #2: Yes

Reviewer #3: Yes

Reviewer #4: Yes

5. Review Comments to the Author

Reviewer #1: The study entitled "Impact of the entorhinal feed-forward connection to the CA3 on hippocampal coding" offers a relevant analysis of neuronal dynamics in hippocampal networks. This is a promising study that adds important insights into hippocampal dynamics. The authors are encouraged to consider the suggestions below to further strengthen the clarity and impact of their findings.

INTRODUCTION

These suggestions should help make the introduction clearer, more structured and engaging, making it easier for readers to understand:

1. Although the introduction contains valuable and detailed information, it can be a bit difficult to follow, especially due to the large volume of information about the functions of the hippocampal subregions and the connections between them. The connection between the different concepts and the objectives of the study could be clearer.

Suggestion: Organize the ideas better, establishing a more fluid narrative line. For example, start with a more general description of the importance of the hippocampus, then introduce the specific functions of each region and, finally, explain the gap that the study aims to fill.

2. The objective of the study is mentioned at the end, but could be more clearly highlighted throughout the introduction. The transition between the historical context and the introduction of the current study could be smoother.

Suggestion: Explain the objective of the study immediately after the description of the functions of the hippocampal regions. For example: "Although the functions of the hippocampal subregions are well understood, the firing dynamics in CA3 in response to DG and EC inputs have not been sufficiently explored. This study aims to investigate how DG and EC inputs affect firing dynamics and interregional communication using a novel microfluidic device design." 3. The introduction makes several references to previous studies and specific functions of the hippocampus, but the way these studies connect to the current study could be more evident. Suggestion: Use more transitional sentences to connect concepts and previous studies. For example: "Although CA3 has a clear role in the rapid encoding of new information, it is still unclear how the communication between the DG and EC affects the firing dynamics in the different hippocampal subregions." 4. The introduction mentions the use of a novel microfluidic device, but there is no clear explanation of how it is used to study hippocampal dynamics. A better explanation of the experimental design would help to contextualize the study. Suggestion: When mentioning the device model, explain a little more about how it works and how it was designed to investigate firing dynamics in hippocampal subregions.

5. The study by Vakilna et al. (2021) is mentioned, but it could be explained more clearly how this study contributes to the basis of your own work.

Suggestion: Explain the role of Vakilna’s study more directly, for example: “In our previous study, Vakilna et al. (2021) demonstrated spontaneous directional firing dynamics in hippocampal subregions, which provided us with a solid basis for exploring how the addition of the EC-CA3 connection affects network dynamics.”

6. The introduction describes the aim and method of the study, but could be more emphatic about the importance of the results for understanding hippocampal function.

Suggestion: Try to include a more emphatic statement about the impact the results may have on the research field.

METHODS

The microfluidic device is described in detail, but the relationship between the various parts and their functionality may be somewhat confusing.

Suggestion: Explain more clearly and concisely how the device was designed to facilitate axonal communication between hippocampal subregions.

DISCUSSION

1. Interpretation of Differences in Firing Patterns: The discussion of the effects of the five-tunnel and four-tunnel architectures on firing activity (feed-forward and feedback) could be further explored. Although differences in firing patterns between these architectures are mentioned, it would be useful to clarify whether these patterns correspond to specific behaviors or cognitive processes, such as memory or spatial navigation.

2. Exclusion of Modulatory Factors: Although the study focused on interactions between hippocampal subregions, it did not consider important modulatory factors such as dopaminergic and cholinergic inputs, which play crucial roles in synaptic plasticity and control of neuronal excitability. The absence of these considerations may limit our understanding of network behavior under more natural conditions. It would be useful to discuss this.

CONCLUSION

The conclusion could be restructured to more clearly emphasize how the findings contribute to our understanding of information processing in the brain.

Reviewer #2: The manuscript provides interesting and potentially valuable information on the effects of layer skipping in neural networks. A point of criticism that, I guess, the authors have met previously is the poor anatomical definition of their system. The findings are presented in the context of the hippocampal tri-synaptic circuit. However, beyond the fact that wells were plated with hippocampal neurons in very roughly approximate densities (parenthetically, the Braitenberg data are dated) found in the hippocampus, no evidence has been presented in this manuscript or the previous papers that were published using similar systems that would show that the quantitative and/or qualitative characteristics of the cell populations are maintained in the system. Hippocampologists will be aware of that, and this manuscript is not the place to remedy the definition of the interconnected regions.

However, the limited number of tunnels connecting the wells and the low number of axons found within them seem to preclude that the quantitative relations of hippocampal interregional connectivities can be maintained in the system used. Connectivities are likely to be much more limited than in vivo.

This point relates to one major point of criticism that I have. The two experimental situations presented do not only differ in the presence of the EC-CA3 pathway, but also in the number of tunnels provided for the classical tri-synaptic loop (51 tunnels without EC-CA3 and 67 tunnels with EC-CA3). I.e. interregional connectivity may have increased by 30%, which, perhaps incidentally, corresponds to the lower bound of changes seen in many of the measures. It is therefore not clear if the observed changes stem from an increase in interregional connectivity or from the layer skipping mediated by direct EC-CA3 connections. This point needs to be addressed in some way. If I have misunderstood this point, the text describing the system needs to be adjusted.

Finally, somewhat at odds with lines 222-229 of the manuscript, there are studies (but, admittedly, only a small part of all hippocampal recording studies) that have done paired-recordings from multiple regions of the hippocampus both in vivo, in acute slices and in organotypic cultures, including acute slices an co-cultures that included the EC. The key advantage of the new device presented here is its potential for ease of access, automation, intervention free monitoring and access to the axon – probably at the cost of many in vivo network properties.

Minor comments

Line 389: please revise sentence

Line 450: twice CA1, please correct

Reviewer #3: This is a noteworthy work that will interest neurophysiologists and computational neuroscientists. However, while the article attempts to compare the 4-tunnel and 5-tunnel architectures, it does not adequately discuss which is better for IBI and ISI. Additionally, there is a lack of justification for some of the methodologies used. Many of the citations used are outdated; approximately half of the articles are over 10 years old. This indicates a need to seek updated information on these topics. Nevertheless, these issues do not detract from the overall soundness of the article.

Reviewer #4: The authors make use of cell culture devices that allow to keep regional-specific neurons in interconnected chambers and dissect the hippocampal connection dynamics. By comparing two designs of the device, a 4-tunnel and a 5-tunnel architecture, they evaluate the lack of the EC-CA3 perforant pathway connection in the spike and burst activity and find higher spike activity with lower burst activity in the 5-tunnel architecture, interpreted as a more "organized" or "fine tuned" activity pattern, which would be allowed by the missing EC-CA3 connection.

The results are interesting, but statistical reporting is lacking, statistical methods used could be improved to better account for the experimental design, and the manuscript possesses inconsistencies and confounds that should be addressed before suitable for publication.

6. PLOS authors have the option to publish the peer review history of their article (what does this mean? ). If published, this will include your full peer review and any attached files.

**Do you want your identity to be public for this peer review?** For information about this choice, including consent withdrawal, please see our Privacy Policy .

Reviewer #1: **Yes: ** Josiane do Nascimento Silva

Reviewer #2: No

Reviewer #3: **Yes: ** Wasiu Balogun

Reviewer #4: **Yes: ** Cesar Coelho

---

## [Author Response · Author response to Decision Letter 1]

27 Mar 2025

Impact of the entorhinal feed-forward connection to the CA3 on hippocampal coding

Response to reviewers

PLOS ONE REQUIREMENTS –

Done

Line 99 now says, “Neonatal mice were exposed to an analgesic ice bath to minimize pain and suffering and then sacrificed by decapitation.”

3. Thank you for stating in your Funding Statement: [The author(s) declare financial support was received for the research, authorship, and/or publication of this article. This study was supported in part by funds from the UC Irvine Foundation. The Foundation had no decision making capacity for the design or manuscript].

Now clearly stated for each author.

4. Thank you for stating the following financial disclosure: [The author(s) declare financial support was received for the research, authorship, and/or publication of this article. This study was supported in part by funds from the UC Irvine Foundation. The Foundation had no decision making capacity for the design or manuscript].

Addressed

5. Thank you for uploading your study's underlying data set. Unfortunately, the repository you have noted in your Data Availability statement does not qualify as an acceptable data repository according to PLOS's standards.

Submitted to Datadryad. This link included in the manuscript will turn will turn active when the paper is accepted. This link can be used by reviewers and http://datadryad.org/share/Y6mWKzH6GE7huPyFQdIRrsGIOsIyL4j3ZPxqu-5hnYU

6. We notice that your supplementary figures are included in the manuscript file. Please remove them and upload them with the file type 'Supporting Information'. Please ensure that each Supporting Information file has a legend listed in the manuscript after the references list.

A supplementary figure list is now included at the end of the manuscript and loaded as required

Reviewer Comments

Reviewer #1: We addressed each of the suggestions below to further strengthen the clarity and impact of their findings.

INTRODUCTION

These suggestions should help make the introduction clearer, more structured and engaging, making it easier for readers to understand:

7. Although the introduction contains valuable and detailed information, it can be a bit difficult to follow, especially due to the large volume of information about the functions of the hippocampal subregions and the connections between them. The connection between the different concepts and the objectives of the study could be clearer.

Suggestion: Organize the ideas better, establishing a more fluid narrative line. For example, start with a more general description of the importance of the hippocampus, then introduce the specific functions of each region and, finally, explain the gap that the study aims to fill.

We responded by moving specifics out of the beginning toward the end and clarified other requested aspects of the introduction in response.

8. The objective of the study is mentioned at the end, but could be more clearly highlighted throughout the introduction. The transition between the historical context and the introduction of the current study could be smoother.

Suggestion: Explain the objective of the study immediately after the description of the functions of the hippocampal regions. For example: "Although the functions of the hippocampal subregions are well understood, the firing dynamics in CA3 in response to DG and EC inputs have not been sufficiently explored.

We fixed this on lines 43-45

9. This study aims to investigate how DG and EC inputs affect firing dynamics and interregional communication using a novel microfluidic device design." 3. The introduction makes several references to previous studies and specific functions of the hippocampus, but the way these studies connect to the current study could be more evident. Suggestion: Use more transitional sentences to connect concepts and previous studies. For example: "Although CA3 has a clear role in the rapid encoding of new information, it is still unclear how the communication between the DG and EC affects the firing dynamics in the different hippocampal subregions." 4. The introduction mentions the use of a novel microfluidic device, but there is no clear explanation of how it is used to study hippocampal dynamics. A better explanation of the experimental design would help to contextualize the study. Suggestion: When mentioning the device model, explain a little more about how it works and how it was designed to investigate firing dynamics in hippocampal subregions.

Our response is on lines 66-68.

10 The study by Vakilna et al. (2021) is mentioned, but it could be explained more clearly how this study contributes to the basis of your own work.

Suggestion: Explain the role of Vakilna’s study more directly, for example: “In our previous study, Vakilna et al. (2021) demonstrated spontaneous directional firing dynamics in hippocampal subregions, which provided us with a solid basis for exploring how the addition of the EC-CA3 connection affects network dynamics.”

We added in last paragraph of introduction (lines 72-77), “In our previous study, Vakilna et al. (2021) demonstrated spontaneous directional spatiotemporal dynamics in soma and axons with the four-tunnel architecture in which revealed classical log-log subregion specific distributed dynamics of interspike intervals (ISIs) and interburst intervals (IBIs) typically found in neural networks (Beggs and Timme, 2012). This provided a solid basis for the exploration of how the additional tunnels connecting EC to CA3 affects network dynamics.”

11 The introduction describes the aim and method of the study, but could be more emphatic about the importance of the results for understanding hippocampal function.

Suggestion: Try to include a more emphatic statement about the impact the results may have on the research field.

Added at the end of introduction (line 80-83), “For the first time, we report comparisons of explicit axonal communication between modeled hippocampal subregions with determined directionality. This could provide important insights for modeling layer skipping in complex network architectures and the subsequent effects on hippocampal coding functions.”

METHODS

12. The microfluidic device is described in detail, but the relationship between the various parts and their functionality may be somewhat confusing. Suggestion: Explain more clearly and concisely how the device was designed to facilitate axonal communication between hippocampal subregions.

Further clarity was added in the paragraph before the first figure (lines 113-116), “The self-wiring nature of the dissociated neuronal cells allows axons to grow through these tunnels. The dimensions of the tunnels are optimized for single axons whose direction of information transmission is captured by the delay in spiking activity across two electrodes in the tunnel.”

DISCUSSION

13 Interpretation of Differences in Firing Patterns: The discussion of the effects of the five-tunnel and four-tunnel architectures on firing activity (feed-forward and feedback) could be further explored. Although differences in firing patterns between these architectures are mentioned, it would be useful to clarify whether these patterns correspond to specific behaviors or cognitive processes, such as memory or spatial navigation.

Added in the section entitled “Function of the EC-CA3 perforant path” (lines 457-466).

14. Exclusion of Modulatory Factors: Although the study focused on interactions between hippocampal subregions, it did not consider important modulatory factors such as dopaminergic and cholinergic inputs, which play crucial roles in synaptic plasticity and control of neuronal excitability. The absence of these considerations may limit our understanding of network behavior under more natural conditions. It would be useful to discuss this.

CONCLUSION

15. The conclusion could be restructured to more clearly emphasize how the findings contribute to our understanding of information processing in the brain.

We wrote a new conclusion.

Reviewer #2: The manuscript provides interesting and potentially valuable information on the effects of layer skipping in neural networks.

16. A point of criticism that, I guess, the authors have met previously is the poor anatomical definition of their system. The findings are presented in the context of the hippocampal tri-synaptic circuit. However, beyond the fact that wells were plated with hippocampal neurons in very roughly approximate densities (parenthetically, the Braitenberg data are dated) found in the hippocampus, no evidence has been presented in this manuscript or the previous papers that were published using similar systems that would show that the quantitative and/or qualitative characteristics of the cell populations are maintained in the system. Hippocampologists will be aware of that, and this manuscript is not the place to remedy the definition of the interconnected regions.

We agree with the reviewer that we did not thoroughly characterize cell types or connections. We apologize for the typo in the statement in methods about cell numbers. On line 98, it said CA3-CA 11:1.25, which now says CA3-CA1 1:1.25. We also added the following and three references in which the ratios are consistent with our methods (lines 98-100) (Braitenberg, 1981; Rapp and Gallagher 1996; Thome et al. 2017; Attili et al. 2022).

17. However, the limited number of tunnels connecting the wells and the low number of axons found within them seem to preclude that the quantitative relations of hippocampal interregional connectivities can be maintained in the system used. Connectivities are likely to be much more limited than in vivo.

Agreed. But this is only a model. In the strictest sense, we could never claim to have a high compliance model of the in vivo hippocampus. But we have learned a lot from slices extricated from their in vivo inputs and a lot from isolated neurons about axonal and dendritic structure and synapses.

18. This point relates to one major point of criticism that I have. The two experimental situations presented do not only differ in the presence of the EC-CA3 pathway, but also in the number of tunnels provided for the classical tri-synaptic loop (51 tunnels without EC-CA3 and 67 tunnels with EC-CA3). I.e. interregional connectivity may have increased by 30%, which, perhaps incidentally, corresponds to the lower bound of changes seen in many of the measures. It is therefore not clear if the observed changes stem from an increase in interregional connectivity or from the layer skipping mediated by direct EC-CA3 connections. This point needs to be addressed in some way. If I have misunderstood this point, the text describing the system needs to be adjusted.

Agreed that the 30% increase in connectivity is exactly the point of the added EC-CA3 connections. We appreciate this observation and have added the following on line 434-437, “Indeed, the additional EC-CA3 layer skipping connection added 30% more axonal connectivity. However, we expect that adding more tunnels alone is unlikely to result in the large changes in spiking and burst behavior that we see due to asymptotic behavior in comparisons of 10, 15 and 51 tunnels (DeMarse et al., 2016).” Again, we have studied a model which has shortcomings like any model. This paper in not the place to debate the usefulness of brain models at every level.

19. Finally, somewhat at odds with lines 222-229 of the manuscript, there are studies (but, admittedly, only a small part of all hippocampal recording studies) that have done paired-recordings from multiple regions of the hippocampus both in vivo, in acute slices and in organotypic cultures, including acute slices an co-cultures that included the EC. The key advantage of the new device presented here is its potential for ease of access, automation, intervention free monitoring and access to the axon – probably at the cost of many in vivo network properties.

We cite two examples of paired whole cell recordings in the hippocampus, lines 233-236, “Efforts to find mono-synaptic connections between somata in two subregions by paired two site intracellular recording in slices are faced with low probabilities, such as 1.3% feedback from CA3 to the dentate (Scharfman 1994) or the need to identify 5-10 CA3 pyramidal cells to sum EPSP’s within 50 ms to excite a single CA1 target (Debanne et al., 1995).”

Minor comments

Line 389 (now 402-403): please revise sentence changed sentence to read, “Differences in burstiness in the feedback pathways were not significant (Fig 6B).”

Line 450: twice CA1, please correct. Removed “CA1 and” on line 447.

Reviewer #3:

20. This is a noteworthy work that will interest neurophysiologists and computational neuroscientists. However, while the article attempts to compare the 4-tunnel and 5-tunnel architectures, it does not adequately discuss which is better for IBI and ISI. Additionally, there is a lack of justification for some of the methodologies used.

We believe that there is a slight misunderstanding about what we mean when we talk about ISI vs IBI. When a comparative adjective like “better” is used, that presupposes an optimal network dynamic state for computation. There is the concept of network criticality which describes in part some of the network dynamics that we are using and how an optimal network should operate, but it is a concept that is not within the scope of this paper. It is impossible for us to measure which type of configuration is better for ISI or IBI as we do not know what an optimal distribution of these would look like. We can only compare these dynamics which go hand-in-hand against the two network architectures we have cultured.

In response to the lack of justification for methodologies used, we are a little confused about which ones you are referring to. Our chosen network dynamics are well defined in this field. Additionally, our culturing methods have been published before. If you are referring to some of the criticisms above about how these networks are not necessarily biologically accurate, please see those comments.

21. Many of the citations used are outdated; approximately half of the articles

---

## [Decision Letter · Decision Letter 1]

PONE-D-24-49403R1Impact of the entorhinal feed-forward connection to the CA3 on hippocampal codingPLOS ONE

Dear Dr. Brewer,

Thank you for submitting your manuscript to PLOS ONE. After careful consideration, we feel that it has merit but does not fully meet PLOS ONE’s publication criteria as it currently stands. Therefore, we invite you to submit a revised version of the manuscript that addresses the points raised during the review process.

We look forward to receiving your revised manuscript.

Kind regards,

Lei An

Academic Editor

PLOS ONE

Reviewers' comments:

Reviewer's Responses to Questions

**Comments to the Author**

1. If the authors have adequately addressed your comments raised in a previous round of review and you feel that this manuscript is now acceptable for publication, you may indicate that here to bypass the “Comments to the Author” section, enter your conflict of interest statement in the “Confidential to Editor” section, and submit your "Accept" recommendation.

Reviewer #1: All comments have been addressed

Reviewer #2: (No Response)

Reviewer #3: All comments have been addressed

Reviewer #4: (No Response)

2. Is the manuscript technically sound, and do the data support the conclusions?

Reviewer #1: Yes

Reviewer #2: Yes

Reviewer #3: Yes

Reviewer #4: Partly

3. Has the statistical analysis been performed appropriately and rigorously? 

Reviewer #1: Yes

Reviewer #2: Yes

Reviewer #3: Yes

Reviewer #4: I Don't Know

4. Have the authors made all data underlying the findings in their manuscript fully available?

Reviewer #1: Yes

Reviewer #2: Yes

Reviewer #3: Yes

Reviewer #4: Yes

5. Is the manuscript presented in an intelligible fashion and written in standard English?

Reviewer #1: Yes

Reviewer #2: Yes

Reviewer #3: Yes

Reviewer #4: Yes

6. Review Comments to the Author

Reviewer #1: After reviewing the manuscript and the authors' responses to previous revisions, I believe that the modifications have been adequately addressed. The methodology and statistical analysis are appropriate, and the data support the conclusions.

The manuscript is clearly presented and written in proper English. No issues related to research ethics or publication ethics were identified.

Reviewer #2: The authors have provided sufficient evidence for the quantitative characteristics of the hippocampal circuitry and clarified the interregional connectivity.

Reviewer #3: Congratulations on a job well done! All of my concerns have been addressed. The hard work and dedication you put into resolving the reviewers' issues deserve commendation. Your attention to detail are truly appreciated.

Reviewer #4: In the first round of reviews, I attached a .txt file with all my comments. Apparently, the authors did not see my file. I apologize for not putting everything directly to the authors. I will mention below what was in the first round and any new comments.

The results are interesting, but statistical reporting is lacking, and the manuscript possesses inconsistencies and confounds that should be addressed before suitable for publication.

Major points

- The statistical results are never properly described. In the Figure legends, the authors mention the statistical tests they used, some R-squared values and some figures have the p-values, but never the actual statistics values, confidence intervals, etc. All statistical results should report the statistic test values, degrees of freedom (if applicable), confidence intervals, p values, and a size effect estimate or a model fit. I suggest summarizing all statistical tests in tables that can be in the supplemental material to avoid polluting the main manuscript, but including them in the publication is essential.

- In Line 245, the subtitle "Spike dynamics of feed-forward and feedback axons reveal greater activity in four-tunnel vs. five-tunnel architecture" seems to be in contradiction with the results, and says the opposite of the conclusion in the following paragraph, (line 270) "This means that spike rates were higher overall in the five-tunnel compared to the four-tunnel configuration.".

- In line 296, the authors say they observe the fastest spiking in EC compared to the other subregions, but the graph 3F shows the DG as the fastest.

- Data and statistical effects visualization could be much improved in the Figures by completely abandoning barplots. They show no information regarding data distribution and often misinform the distribution in certain scales. Showing the data points or a violin plot along with the mean +- sd/se, for instance, would allow readers to better visualize the data distribution and mean.

- From the methods, it is not clear if these neuronal network devices allow for connections not present in the brain (ex., CA3 - EC feedforward connections). If Yes, then

1) The authors should mention it.

2) Also if yes, this leads to a possibly better experimental control from a network science perspective. In network science, the fact that the 4-tunnel architecture has less routes of interaction could per se account for some of the observed changes here and many changes in most networks (biological or not). Thus an experimental control with the same possible interaction routes would be more appropriate. Although a 5-tunnel architecture with a "wrong connection" is beyond the scope of this study, it could show precisely how, mechanistically, the tri-synaptic design is biologically more efficient. If artificial connections are possible in this method, this possibility of experimental control should be discussed.

- The names of the architectures vary in many ways across the whole manuscript (eg. 5-tunnel and 4-tunnel and four-tunnel and five-tnnel) and should be standardized.

Minor points

- The authors could gain much leverage from using statistical models that better fit their experimental design, such as linear mixed models with random effects on the plate. such models also allow for generalizations to other distributions beyond gaussian (eg. Poisson, gamma, etc).

- Authors should update the link where the data and code are available since the given one seems to not exist or has been changed.

- In SupFig1, panels C-D, if the graphs match the colors of the dots OR the curves to that of the legend, it would help a more rapid identification.

- The "legend titles" should all be reviewed and corrected. They show several inconsistencies. For example, in Fig 2 the legend is missing the 5-tunnel architecture name. In Fig 5, the legend title says the opposite of what the results say. In Fig 6, the legend title does not seem to make sense.

- In lines 304-308,

"Comparison of the two tunnel types in Fig 3G, shows only a 4% higher spike rate in EC for the five-connect networks, while all the others produced an 11-80% advantage for the four-tunnel networks over those of the five-connect.". However, in Fig 3G, the difference magnitude between the EC bars seems comparable to the largest difference in other subregions (that of CA1).

- The bar graphs show p-values and deltas in some figures but not in others containing similar comparisons. Showing those per comparison along with the deltas mentioned in the main text will facilitate matching text and graph and reveal the issue in the previous comment.

7. PLOS authors have the option to publish the peer review history of their article (what does this mean? ). If published, this will include your full peer review and any attached files.

**Do you want your identity to be public for this peer review?** For information about this choice, including consent withdrawal, please see our Privacy Policy .

Reviewer #1: No

Reviewer #2: No

Reviewer #3: No

Reviewer #4: **Yes: ** Cesar Coelho

---

## [Author Response · Author response to Decision Letter 2]

13 May 2025

Impact of the entorhinal feed-forward connection to the CA3 on hippocampal coding

Response to Reviewers 250512

Reviewer Comments

Reviewer #4: In the first round of reviews, I attached a .txt file with all my comments. Apparently, the authors did not see my file. I apologize for not putting everything directly to the authors. I will mention below what was in the first round and any new comments.

The results are interesting, but statistical reporting is lacking, and the manuscript possesses inconsistencies and confounds that should be addressed before suitable for publication.

Major points

1. The statistical results are never properly described. In the Figure legends, the authors mention the statistical tests they used, some R-squared values and some figures have the p-values, but never the actual statistics values, confidence intervals, etc. All statistical results should report the statistic test values, degrees of freedom (if applicable), confidence intervals, p values, and a size effect estimate or a model fit. I suggest summarizing all statistical tests in tables that can be in the supplemental material to avoid polluting the main manuscript, but including them in the publication is essential.

We agree that effect sizes are important. They are now included in results as partial eta-squared (η²) values. Appendix tables along with statistic test values, degrees of freedom (if applicable), p values, and a partial η² estimates or an R2 model fit are now added for each figure.

2. In Line 245, the subtitle "Spike dynamics of feed-forward and feedback axons reveal greater activity in four-tunnel vs. five-tunnel architecture" seems to be in contradiction with the results, and says the opposite of the conclusion in the following paragraph, (line 270) "This means that spike rates were higher overall in the five-tunnel compared to the four-tunnel configuration."

Line 245 – now says “Dynamics of feed-forward and feedback axons reveal faster spiking in five-tunnel vs. four-tunnel architecture”. This now agrees with lines 271-272 at the end of the paragraph, “slopes in the five-tunnel design were 30-87 % faster than the four-tunnel axon architectures.

3. In line 296, the authors say they observe the fastest spiking in EC compared to the other subregions, but the graph 3F shows the DG as the fastest.

We found an error in our 3F bar graph. The current version matches the curves shown in 3A-D. Line 308 now says, “the fastest spiking in EC compared to the other subregions (Fig 3F).”

4. Data and statistical effects visualization could be much improved in the Figures by completely abandoning barplots. They show no information regarding data distribution and often misinform the distribution in certain scales. Showing the data points or a violin plot along with the mean +- sd/se, for instance, would allow readers to better visualize the data distribution and mean.

We believe the underlying data distribution is adequately displayed in the log-log cumulative probability distributions described in detail in lines 186-209. The bar plots are a summary of the slopes of the data distribution in the log-log CCF. Data distributions are most useful to validate linear Gaussian distributions about a mean. It would be foolish to do this for 4,000 to 35,000 ISI data points that are log-log distributed over more than an order of magnitude. To see the variation and confidence of the slope from the mean linear fit in the log domain (equation 2 on line 193), we have added R2 goodness of fit and other details of the ANCOVA in the supplementary tables.

Here is an example of the data distribution and the resultant CCF that results in curve smoothing.

5. From the methods, it is not clear if these neuronal network devices allow for connections not present in the brain (ex., CA3 - EC feedforward connections). If Yes, then

1) The authors should mention it. Now mentioned in the Limitations section of the Discussion.

2) Also if yes, this leads to a possibly better experimental control from a network science perspective. In network science, the fact that the 4-tunnel architecture has less routes of interaction could per se account for some of the observed changes here and many changes in most networks (biological or not). Thus an experimental control with the same possible interaction routes would be more appropriate. Although a 5-tunnel architecture with a "wrong connection" is beyond the scope of this study, it could show precisely how, mechanistically, the tri-synaptic design is biologically more efficient. If artificial connections are possible in this method, this possibility of experimental control should be discussed.

On line 556, we add the limitation in our network “the observation of feedback activity from CA3 to EC (not shown) indicates that some unphysiological connections are also possible when evidence for this connection is lacking in vivo (Ishizuka et al., 1990).

Ishizuka N, Weber J, Amaral DG (1990) Organization of intrahippocampal projections originating from CA3 pyramidal cells in the rat. J Comp Neurol 295:580-623.10.1002/cne.902950407

6. The names of the architectures vary in many ways across the whole manuscript (eg. 5-tunnel and 4-tunnel and four-tunnel and five-tunnel) and should be standardized.

Naming consistency is now four-tunnel and five-tunnel throughout, except for the spatially constrained figures.

Minor points

7. The authors could gain much leverage from using statistical models that better fit their experimental design, such as linear mixed models with random effects on the plate. such models also allow for generalizations to other distributions beyond gaussian (eg. Poisson, gamma, etc).

We do have categorical differences that we are most interested in……e.g. slopes for the log-log distribution of interspike intervals (ISI) between four-tunnel vs. five-tunnel network architectures. And we used ANCOVA to test whether the two architectures are likely to result from two different populations or whether their variances are likely to result from subsamples of a single distribution. The detailed statistics are now included in supplemental tables.

8. Authors should update the link where the data and code are available since the given one seems to not exist or has been changed.

The problem with Dryad, has been fixed.

9. In SupFig1, panels C-D, if the graphs match the colors of the dots OR the curves to that of the legend, it would help a more rapid identification.

The curves are now adequately identified by the colors of the R2 values and names

10. The "legend titles" should all be reviewed and corrected. They show several inconsistencies. For example, in Fig 2 the legend is missing the 5-tunnel architecture name. In Fig 5, the legend title says the opposite of what the results say. In Fig 6, the legend title does not seem to make sense.

Fig 2 now addressed in a revised figure

Fig 5 In line 390, we replaced faster with slower

11. In lines 304-308, "Comparison of the two tunnel types in Fig 3G, shows only a 4% higher spike rate in EC for the five-connect networks, while all the others produced an 11-80% advantage for the four-tunnel networks over those of the five-connect.". However, in Fig 3G, the difference magnitude between the EC bars seems comparable to the largest difference in other subregions (that of CA1).

Sorry, but we don’t understand “seems comparable to the largest difference in other subregions (that of CA1).” The difference in CA1 appears as an 80% decrease in slope for the five- relative to the four-tunnel, compared to the stated 4% increase for EC.

12.The bar graphs show p-values and deltas in some figures but not in others containing similar comparisons. Showing those per comparison along with the deltas mentioned in the main text will facilitate matching text and graph and reveal the issue in the previous comment.

Our figures from the last revision have fixed these problems.

---

## [Decision Letter · Decision Letter 2]

Impact of the entorhinal feed-forward connection to the CA3 on hippocampal coding

PONE-D-24-49403R2

Dear Dr. Gregory J Brewer,

We’re pleased to inform you that your manuscript has been judged scientifically suitable for publication and will be formally accepted for publication once it meets all outstanding technical requirements.

Kind regards,

Lei An

Academic Editor

PLOS ONE

**Comments to the Author**

1. If the authors have adequately addressed your comments raised in a previous round of review and you feel that this manuscript is now acceptable for publication, you may indicate that here to bypass the “Comments to the Author” section, enter your conflict of interest statement in the “Confidential to Editor” section, and submit your "Accept" recommendation.

Reviewer #4: All comments have been addressed

2. Is the manuscript technically sound, and do the data support the conclusions?

Reviewer #4: Yes

3. Has the statistical analysis been performed appropriately and rigorously? 

Reviewer #4: Yes

4. Have the authors made all data underlying the findings in their manuscript fully available?

Reviewer #4: Yes

5. Is the manuscript presented in an intelligible fashion and written in standard English?

Reviewer #4: (No Response)

6. Review Comments to the Author

Reviewer #4: (No Response)

---

## [Editor Report · Acceptance letter]

PONE-D-24-49403R2

PLOS ONE

Dear Dr. Brewer,

I'm pleased to inform you that your manuscript has been deemed suitable for publication in PLOS ONE. Congratulations! Your manuscript is now being handed over to our production team.

Kind regards,

on behalf of

Dr. Lei An

Academic Editor

PLOS ONE